# Two Consecutive Prolines in the Fusion Peptide of Murine β-Coronavirus Spike Protein Predominantly Determine Fusogenicity and May Be Essential but Not Sufficient to Cause Demyelination

**DOI:** 10.3390/v14040834

**Published:** 2022-04-17

**Authors:** Abass Alao Safiriyu, Manmeet Singh, Abhinoy Kishore, Vaishali Mulchandani, Dibyajyoti Maity, Amrutamaya Behera, Bidisha Sinha, Debnath Pal, Jayasri Das Sarma

**Affiliations:** 1Department of Biological Sciences, Indian Institute of Science Education and Research Kolkata, Mohanpur 741246, India; saa19rs001@iiserkol.ac.in (A.A.S.); manmeetsinghbiotech@gmail.com (M.S.); abhinoy@gmail.com (A.K.); vm17ip003@iiserkol.ac.in (V.M.); ab16m081@iiserkol.ac.in (A.B.); bidisha.sinha@iiserkol.ac.in (B.S.); 2Department of Computational and Data Sciences, Indian Institute of Science, Bengaluru 560012, India; djmaity@gmail.com (D.M.); dpal@iisc.ac.in (D.P.)

**Keywords:** β-Coronavirus, mouse hepatitis virus-A59/MHV-A59, mouse hepatitis virus-2/MHV2 spike protein, fusion peptide/FP, cell-to-cell fusion (fusogenicity), neuropathogenesis, hepatitis, demyelination, structural rigidity

## Abstract

Combined in silico, in vitro, and in vivo comparative studies between isogenic-recombinant Mouse-Hepatitis-Virus-RSA59 and its proline deletion mutant, revealed a remarkable contribution of centrally located two consecutive prolines (PP) from Spike protein fusion peptide (FP) in enhancing virus fusogenic and hepato-neuropathogenic potential. To deepen our understanding of the underlying factors, we extend our studies to a non-fusogenic parental virus strain RSMHV2 (P) with a single proline in the FP and its proline inserted mutant, RSMHV2 (PP). Comparative in vitro and in vivo studies between virus strains RSA59(PP), RSMHV2 (P), and RSMHV2 (PP) in the FP demonstrate that the insertion of one proline significantly resulted in enhancing the virus fusogenicity, spread, and consecutive neuropathogenesis. Computational studies suggest that the central PP in Spike FP induces a locally ordered, compact, and rigid structure of the Spike protein in RSMHV2 (PP) compared to RSMHV2 (P), but globally the Spike S2-domain is akin to the parental strain RSA59(PP), the latter being the most flexible showing two potential wells in the energy landscape as observed from the molecular dynamics studies. The critical location of two central prolines of the FP is essential for fusogenicity and pathogenesis making it a potential site for designing antiviral.

## 1. Introduction

Fusion peptide (FP) is one of the functional segments of virus-host attachment spike protein that is important for viral fusion to either host cell plasma membrane or endosomal membrane [1,2]. During the initial stage of the fusion cascade, spike protein unfolds and extends to expose FP to the target membrane for anchoring via the fusion core domain [3,4]. Mounting evidence from a large body of literature highlights that FP is a valuable target for the development of pan-CoV therapeutics owing to its conserved nature and contribution to mediating the fusion of the viral and host cell membrane, and driving its fusion mechanism across the coronavirus (CoV) family [5,6,7,8,9,10,11,12]. The ongoing COVID-19 pandemic highlights the immediate requirement to develop effective therapeutics against the existing human-β-coronavirus, SARS-CoV-2, and other infectious coronavirus strains, or those emerging in the near future [13,14,15]. While a lot of work has been done on SARS-CoV-2, its highly infectious nature limits its regular laboratory use. Limited experimental evidence, therefore, exists for human β-coronavirus, but alternate evidence from prototype murine-β-coronavirus may shed some light on the understanding of the intricate mechanism of the FP-mediated fusion process and its associated pathogenesis [16,17].

Two very closely related strains of murine-β-coronavirus MHV-A59 and MHV2 differ in their fusogenic properties and hepato-neuropathogenesis [18,19,20,21,22]. MHV-A59 is highly fusogenic and causes acute hepatitis, meningoencephalomyelitis with chronic progressive demyelination concurrent with axonal loss and is denoted as a neurotrophic strain [22,23]. In contrast, non-neurotropic strain MHV-2 sharing 91% genome identity with MHV-A59, and 83% pairwise spike sequence identity, causes only meningitis and is unable to invade the brain parenchyma. A series of detailed comparative in vitro and in vivo studies previously demonstrated that spike protein is one of the major determinants of cell-to-cell fusion, viral infectivity, viral antigen spread, and its consequent neuropathology, demyelination, and axonal loss [18,19,21,24]. Target RNA recombination using a reverse genetic system engineering two isogenic spike gene recombinant strains of MHV, RSA59, and RSMHV2 have given interesting insight. The spike gene recombinant strain of MHV, RSA59 where the spike gene was replaced from parental neurotropic and demyelinating strain MHV-A59, and RSMHV2 where the spike gene was derived from parent hepatotropic non-demyelinating strain MHV2 share the same genetic background except for the spike gene [21,25]. Both RSA59 and RSMHV2 can efficiently infect neurons but RSA59 can spread from neuron to neuron more specifically from gray matter to white matter following axonal transport and can release at the nerve end to directly infect oligodendrocytes in the white matter through cell-to-cell fusion [26,27]. Thus, RSA59 can evade the immune system, silently infect white matter oligodendrocytes, and cause moderate to severe demyelination. In contrast, RSMHV2 as mentioned earlier can infect the neuron but is impaired in axonal transport and cannot reach the white matter oligodendrocytes. Impaired axonal transport and lack of cell-to-cell fusion properties contribute to impaired demyelination in RSMHV2. Previous studies have demonstrated that RSA59 can reach the optic nerve via retrograde axonal transport and cause optic neuritis including neuron inflammation, demyelination, and axonal loss [24,28]. In contrast, RSMHV2 was unable to follow retrograde transport and thus was unable to induce optic neuritis. Retrograde axonal transport of RSA59 is also known to damage retinal ganglionic cells as a consequence of cell-to-cell fusion, whereas RSMHV2 is impaired in causing retinal ganglionic cell loss due to lack of retrograde axonal transport and infection to retinal ganglionic cells [24,28]. Another seminal study combining exogenous spike protein trafficking as well as in vivo and in vitro viral spread and dissemination demonstrated that irrespective of the presence of known murine-coronavirus viral entry receptor CEACAM1, spike protein by itself can initiate the fusion process [17,29]. Different strains of CoV induce differential cell-to-cell fusion and are responsible for the differential pathogenicity and disease severity. In summary, the spike protein plays a vital role in CoV-induced cell-to-cell fusion and pathogenesis. To delineate the minimum essential motif required for fusogenicity, studies were targeted to the FP of the S2 domain of the spike protein owing to its key role in early events in cell-to-cell fusion necessary for intercellular viral spread. Two consecutive central proline residues in the FP domain were identified to play a crucial role in the event of the fusion process. In one of the previous studies, it has been demonstrated by generating a proline deletion mutant that a proline deletion or insertion in the FP may significantly alter murine CoV(m-CoV) induced fusogenicity, viral antigen spread, infectivity, and its consequent neuropathological event, demyelination, and axonal loss [17].

We had previously demonstrated that the deletion of single proline from the centrally located double prolines in the FP of fusogenic and demyelinating strain RSA59 (PP) led to the loss of fusogenicity due to the loss of structural rigidity around the FP neighborhood from the lack of double prolines [16,17]. In this study, we add one proline at the center of the FP of fusion impaired non-demyelinating strain RSMHV2 (P) and investigate whether only two consecutive prolines with the parental neighboring amino acids within a membrane environment are sufficient to provide the rigidity required for syncytia formation. We, therefore, generated a mutant RSMHV2 (PP) in which one proline was added to RSMHV2 (P) and compared with RSA59 (PP) and RSMHV2 (P). Comparative studies reveal that RSMHV2 (PP) can induce cell-to-cell fusion, spread through the neuron, and can occasionally reach the white matter and thus can induce discrete myelin loss, but is unable to form a concentrated demyelinating plaque. An increased ability to cause cell-to-cell fusion, viral antigen spread, and consecutive myelin loss could be due to the increased rigid structure induced by one additional proline in the FP neighborhood but it was restricted as compared to RSA59 (PP). Previous in silico studies had identified the cause to be the transformation of the FP structure from helix-turn-helix-turn-helix in RSA59 (PP) to a helix-loop-helix in RSA59 (P). We find the same in this study from the comparison of RSMHV2 (PP) versus RSMHV2 (P). With the local torsional flexibility, the secondary structure of RSMHV2 (PP) is identical to RSA59 (PP). Two prolines are required for efficient viral spread from the brain to the spinal cord and from the brain to the retina via the optic nerve to the eye by retrograde axonal transport. Altered rigidity of the FP may contribute significantly to cell-to-cell fusion and viral spread which may help in designing the therapeutic strategies for anti-pan-CoV.

## 2. Materials and Methods

### 2.1. Viruses

A recombinant isogenic demyelinating strain of RSA59 (PP) and non-demyelinating strain RSMHV2 (P) expressing enhanced-Green Fluorescent Protein (EGFP) have been used for understanding spike mediated fusogenicity and hepato-neuropathogenesis in our series of previous studies [25,26,30]. Briefly, RSMHV2 (PP) expressing EGFP was engineered by inserting one additional proline through quick-change site-directed mutagenesis combined with targeted RNA recombination next to the existing central proline of the parental RSMHV2 (P) FP of spike protein [21,31]. The plasmids were isolated and linearized. Synthetic RNA was obtained using an in vitro transcription kit. The synthetic capped RNA as a donor and fMHV as a recipient virus were subjected to targeted RNA recombination methods [25]. The resultant recombinant strains were selected and sequenced for verification. Overall, the viruses are isogenic recombinant strains of MHV-A59 expressing EGFP but differ in spike gene, where RSA59 (PP) possesses spike gene of the parental strain of MHV-A59, RSMHV2 (P) possesses spike gene of the fusion impaired non-demyelinating parental strain of MHV2 and mutant RSMHV2 (PP) also possesses the spike gene of the fusion impaired non-demyelinating parental strain of MHV2 with one proline insertion in the FP.

### 2.2. Isolation and Enrichment of Primary Neuron from Neonatal Day 0 Mouse Brain

Whole brains were harvested from day 0 pups and meninges were carefully removed from brain tissue. Homogenized brain tissues were incubated in a rocking water bath set at 37 °C for 30 min in Hanks’ balanced salt solution (Gibco), containing 300 μg/mL DNase I (Sigma, Tokyo, Japan) and 0.25% trypsin (Sigma). Enzyme-dissociated cells were triturated in the presence of 0.25% FBS, followed by a wash and centrifugation (300× *g* for 10 min). The pellet was again resuspended in Hanks’ balanced salt solution and passed through a 70-μm nylon mesh. A second wash and centrifugation (300× *g* for 10 min) were performed, and finally, the cell pellet was diluted to 10^6^ cells/mL with (DMEM) containing 1% HEPES, 1% penicillin, 1% streptomycin, 1% non-essential amino acid, 1% L-glutamine, 10% FBS. Cells were plated on PolyDiLysin and Laminin (PDL/Lam) coated culture plates and allowed to adhere for 1 day in a humidified CO_2_ incubator at 37 °C. After 24 h, all non-adherent cells were removed, and cells were switched to a serum-free, growth medium (Neurobasal medium containing B27). The purity of isolated neuronal cells was determined by double-label immunofluorescence with neuronal marker anti-MAP2 and astrocyte marker anti-GFAP (glial fibrillary acidic protein).

### 2.3. Maintenance of Secondary Cells in Culture

L2 is a murine lung epithelial cell line (CCL-149) and Neuro2a is a mouse neuroblastoma cell line (CCL-131) that was obtained from the American Tissue Culture Collection (ATCC). DBT is a mouse astrocytic delayed brain tumor cell line (JCRB1580) that was obtained from the Japanese Collection of Research Bioresources Cell Bank. Neuro2A cells were cultured in Minimum Essential Medium (MEM) containing 10% FBS and DMEM containing 7.5% NaHCO_3_, 1M HEPES, 1% penicillin, 1% streptomycin, 1% non-essential amino acid, 1% L-glutamine, 10% FBS was used to maintain for L2 and DBT cells. All the cells were maintained in continuous culture up to 5 passages for experimentation. Cells were cultured on etched coverslips as per requirement.

### 2.4. Viral Infection in Primary Neuronal Cells, Lung Epithelial Cells L2, Neuroblastoma Cells Neuro2A, and Astrocytoma Cells DBT

Confluent monolayers of primary neurons were infected with 2 MOI and, L2, Neuro2A cells, and DBT cells were infected with 0.5 MOI of RSA59 (PP), RSMHV2 (P), and RSMHV2 (PP) inoculum prepared in respective media containing 2% FBS. Infected cells were incubated with the virus at 37 °C in a humidified incubator with 5% CO_2_ for 1 h 15 min with intermittent rocking at 15 min intervals for efficient viral adsorption. The viral inoculum was discarded and the infected cells were cultured with their respective growth medium in presence of 5% CO_2_ at 37 °C. After 8 h, 12 h, 16 h, and 24 h post-infection infected cells were either processed for studying viral antigen spread and syncytia formation by EGFP fluorescence or immunofluorescence colocalizing viral antigen with different neuroglial cell-specific markers.

### 2.5. Immunofluorescence on Cultured Cells

Infected cells were washed with 1X PBS containing Ca^2+^ and Mg^2+^, fixed in 4% PFA, washed with 1X PBS, and then mounted on glass slides using Mowiol 4–88 mounting media with DAPI. For neuron enrich primary culture characterization, cells were permeabilized with Triton X-100 in PBS. The cells were incubated with a blocking solution of PBS containing Triton X-100 and goat serum (GS). The cells were then incubated with primary antibodies rabbit anti-MAP2 (neuron marker), mouse anti-NFM (neuron marker), mouse anti-GFAP (astrocyte marker), at dilution 1:200 and mouse anti-H8H9 (same as anti-GalC, matured oligodendrocyte marker) at dilution 1:50 prepared in PBS/GS/Triton X-100 for 1 h. The antibodies used with their source and dilutions are tabulated in Table 1. The primary antibody labeled cells were then washed thrice with 1X PBS for 5 min and then incubated with respective secondary antibodies prepared in PBS/GS/Triton X-100 for 1 h, as specified in Table 1. The fluorescence-labeled cells were carefully washed with PBS carefully avoiding exposure to light and then mounted. The slides were then observed for EGFP fluorescence. Images were acquired with a Nikon eclipse Ti2 epifluorescence microscope with Nikon DS-Qi2 coupled camera and analyzed using ImageJ software.

### 2.6. Quantification of RSA59 (PP), RSMHV2 (P), and RSMHV2 (PP) Induced Cell-to-Cell Fusion In Vitro

The merged images of EGFP and DAPI channels were used for the quantification. The mean nuclei per syncytium formed in Neuro2A cells were quantified by counting the number of DAPI-stained nuclei inside the EGFP expressing syncytia. A total of 30 frames of 20× images were taken from three independent experiments for every time point for each strain of the virus. Randomly selected 25 syncytia were considered for precise quantification of nuclei stained with DAPI within a syncytium and graphically presented for L2 and Neuro2A cells. As the syncytia formed by RSA59 (PP) in DBT were very big, the size of each nucleus was determined and used to calculate the number of nuclei in each syncytium by using software algorithms. The syncytia formed by RSMHV2 (P) and RSMHV2 (PP) were not distinct, thus clusters of infected cells were considered as syncytia for DBT cells. Like in Neuro2A, each cluster was cropped out and the nuclei from the DAPI channel was counted. ImageJ 1.52 g (Fiji) software was used for quantification.

### 2.7. Inoculation of Mice

Four-week-old MHV-free C57BL/6J mice were intracranially inoculated with a 50% LD50 dose of 20,000 PFU/mL RSA59 (PP), RSMHV2 (PP), or 100 PFU/mL RSMHV2 (P) as previously described [16,30]. Mice were monitored for the signs and symptoms of disease and possible mortality. Mock-infected groups received PBS with 0.75% BSA and were housed in the same conditions as infected groups. On days 3, 5, and 6 (acute stage) and day 30 (chronic stage) p.i., mice were anesthetized, transcardially perfused with PBS followed by 4% PFA, and successively the brains, spinal cords, and livers were harvested. Six mice (N = 6 in two experiments) were inoculated in each infection group for histopathological analysis on days 3, 6, and 30. Three mice (N = 3) were used for each infection group alongside mock-infected mice for immunofluorescence analysis on day 5 p.i. spinal cord cryosections.

### 2.8. Histopathological Analyses

Mice were sacrificed at days 3, 6, and 30 p.i. and perfused transcardially with PBS followed by 4% paraformaldehyde (PFA) in PBS. Liver, brain, and spinal cord tissues were harvested, postfixed in 4% PFA overnight, tissues were transferred to 70% ethanol and processed for tissue routine paraffin sectioning. Five μm thick longitudinal liver sections, sagittal brain sections, and cross-sections of spinal cord tissues were prepared for further histological and immunohistochemical analysis. Liver and spinal cord (cervical, thoracic and lumbar) tissue sections were processed for H&E and Luxol Fast Blue (LFB) staining, respectively. Liver and spinal cord pathology were blindly analyzed by two other investigators.

### 2.9. Immunohistochemical Analysis and Quantification of Viral Antigen

Serial sagittal sections of the brain and cross-sections of spinal cord tissues were stained using the avidin-biotin-immuno-peroxidase technique (Vector Laboratories, Burlingame, CA, USA) with 3, 3′ diaminobenzidine (DAB) as substrate and anti-N antibody as primary antibody, which was a kind gift from Dr. Julian Leibowitz of Texas A&M, College Station, Texas. The degree of viral antigen staining in different neuroanatomic regions of infected mice brain tissues was evaluated based on the scoring scale; score 0: no apparent viral antigen staining; 1: very small foci of viral antigen-positive cells; 2: widespread but small foci of viral antigen-positive cells; 3: widespread large foci of viral antigen-positive cells. To quantify anti-viral staining in the spinal cord, Fiji (ImageJ 1.52 g) software was used [32]. Image analysis was performed using the basic densitometric thresholding features of Fiji. The image was first captured at 10× magnification which allowed the entire section of the spinal cord to be visualized within a single image frame. The RGB image was color-deconvoluted into three different colors to separate the DAB-specific staining. The background labeling was also subtracted from all images, and then the contrast was slightly enhanced to improve the resolution. The perimeter of the spinal cord was digitally outlined, and the area was calculated in µm^2^. A threshold value was defined for each image. The magnitude of viral staining was defined as the percentage area of staining (ratio of target-stained area to total selected area multiplied by 100), as previously described [17]. Two other investigators blindly analyzed all the sections following the above-described method.

### 2.10. Immunofluorescence on Spinal Cord Cryosections

Spinal cord tissues harvested on day 5 p.i.were fixed in PBS containing 4% PFA for 8 h and then placed in PBS containing 10% sucrose at 4 °C overnight and subsequently transferred to PBS containing 30% at 4 °C for 24 h. Tissues were embedded in OCT medium (cryomatrix) and coronally sectioned at 8µm with the aid of a cryotome (Thermo Scientific, Waltham, MA, USA). Sectioned tissues were placed on the charged slides and kept at −80 °C till the time of immunofluorescence. Frozen sections were immunofluorescently labeled as previously described [27,30,33]. Briefly, sections were postfixed with ice-cold 95% ethanol for 20 min and washed with PBS at room temperature for 10 min followed by incubation with 1M glycine in PBS for 1 h to reduce nonspecific cross-linking, and 1 mg/mL NaBH4 for 10 min to decrease autofluorescence at room temperature in a humidified chamber. The sections were again PBS washed thrice for 5 min each before being incubated with a blocking solution containing 2.5% goat serum in PBS (GS) and 0.5% triton-X-100 in PBS for 1 h at room temperature. Incubation of sections with primary antibodies; virus-specific anti-nucleocapsid (anti-N) and anti-MAP2 prepared in GS at 1:50 and 1:200 dilutions respectively were done overnight at 4 °C. The section was then PBS washed thrice for 5 min before incubation with secondary antibodies as mentioned in Table 1. All incubations were carried out in a humidified chamber. Tissue sections were then washed with PBS thrice for 5 min and subsequently mounted with Mowiol 4–88 mounting medium containing DAPI. A green channel was used to visualize the viral antigen. MAP2 staining was viewed in the red channel. Visualization and images were acquired with a Nikon eclipse Ti2 epifluorescence microscope with Nikon DS-Qi2 coupled camera and processed with Image J (Fiji) software.

**Table 1 viruses-14-00834-t001:** Antibodies and respective dilutions used for immunofluorescence.

Primary Antibody	Dilution	Secondary Antibody	Dilution
Rabbit polyclonal Anti-MAP2 (Sigma)	1:200	Alexa fluor 568 Donkey Anti Rabbit (Invitrogen)	1:800
Mouse monoclonal Anti-NFM (Sigma)	1:200	FITC Goat Anti Mouse IgG (Jackson immunoresearch, West Grove, PA, USA)	1:250
Mouse monoclonal Anti-GFAP (Sigma)	1:200	FITC Goat Anti Mouse IgG (Jackson immunoresearch, West Grove, PA, USA)	1:250
Anti-H8H9 (same as anti-Gal C) (mouse monoclonal antibody against mature Oligodendrocytes) [34]	1:50	FITC Goat Anti Mouse IgG (Jackson immunoresearch, West Grove, PA, USA)	1:250
Anti-N (anti-nucleocapsid of MHV) (Gift from Dr. Julian Leibowitz of Texas A&M, College Station, TX)	1:25	FITC Goat Anti Mouse IgG (Jackson immunoresearch, West Grove, PA, USA)	1:250

### 2.11. Detection of Demyelination and Quantification

Spinal cord (cervical, thoracic and lumbar) tissue sections were processed for Luxol Fast Blue (LFB) staining to detect myelin loss. Total white matter area and areas with myelin loss (marked by a plaque of no LFB stain within white matter) were determined on day 30 RSA59 (PP), RSMHV2 (P), and RSMHV2 (PP) p.i. mice. Four-six LFB-stained spinal cord cross-sections from each mouse were randomly selected and analyzed using Fiji software (ImageJ 1.52 g) [17]. The total number of mice in each group was 5–6. The total perimeter of the white matter regions in each cross-section was outlined and calculated by adding up the dorsal and ventrolateral white matter areas in each section. The total perimeter of the demyelinated regions was also outlined and added for each section separately. The percentage of spinal cord demyelination per section per mouse was obtained by dividing the total area of the demyelinating plaque over the total area of the calculated white matter and then multiplied by 100, as previously described [17].

### 2.12. Structure Modeling and Molecular Dynamics Simulation of RSMHV2 (PP)

The structure of the trimeric fusion domain of RSMHV2 (PP) was obtained through homology modeling using MODELLER with RSA59 (PP) as the template [35]. All-atom MD simulation of this modeled structure was performed with GROMACS 5.1.4 using the CHARMM force field with cmap following the protocol previously described [17]. The simulated region of the RSMHV2 (PP) fusion domain, residues 909–1154 of each chain, corresponds to the same region of RSMHV2 (P) previously studied. The structure was centered in a cubic box with a dimension chosen by GROMACS to ensure a distance of 1 nm between the protein edge and box face. The system was solvated with water. Na^+^ and Cl^−^ ions were added to neutralize the charge on the protein and ensure a final concentration of 0.1 M. Periodic boundary condition in all directions was applied. The system was energy minimized with the steepest descent algorithm, and the minimization was stopped when the maximum force in the system dropped below 1000 kj mol^−1^ nm^−1^. This was followed by equilibration under NVT and NPT ensembles for 2 ns each. Berendsen thermostat at a temperature of 300 K and Parrinello-Rahman barostat at 1 atm was used in the simulations [36,37]. The production MD was run for 500 ns with a time step of 2 fs with frames saved every 10 ps yielding 50,000 frames for analysis.

The comparative analysis of this MD trajectory with the previous simulations [17] was done with MD DaVis, an analysis tool developed in-house (https://github.com/djmaity/md-davis, accessed on 18 January 2022). First, the free energy landscapes were calculated using root mean square deviation and radius of gyration following the protocol of Tavernelli et al., 2003 [38]. The free energy landscapes of the full trajectory showed that the systems were still evolving for the first 100 ns. Therefore, the trajectory from 100–500 ns was used for replotting the free energy landscapes and subsequent analysis. Next, the secondary structure was calculated with the GROMACS do_dssp tool. The percentage of frames for which a specific secondary structure was observed was calculated at each residue location. The torsional flexibility was also calculated, defined as the circular standard deviation of the φ and ψ backbone torsional angles. Finally, the hydrogen bonds with the central prolines in the FP were calculated with the GROMACS hbond tool and the percentage of frames with the hydrogen bonds was tabulated.

### 2.13. Statistical Analysis

The data from syncytia formation in the cultured cells was computed and analyzed as described above. Unpaired Student’s *t*-test was used to compare two strains of virus at each time point p.i. The level of significance for viral antigen and LFB staining in different neuroanatomic regions and spinal cord sections were calculated using one-way ANOVA followed by Tukey’s multiple comparisons test. All data were plotted and analyzed using GraphPad Prism 6.01 software. The level of significance and the means are presented in a scatter-bar diagram. A *p*-value < 0.05 was considered statistically significant and presented as * *p*. The standard error of the mean (SEM) was presented with each mean as an error bar in the plots.

## 3. Results

### 3.1. Sequence Comparison between RSA59 (PP), RSMHV2 (P), RSMHV2 (PP) Fusion Peptide

The mutant RSMHV2 (PP) was generated by adding one proline to FP by site-directed mutagenesis and targeted RNA recombination as discussed in the “Material and Methods”. The entire FP of RSA59 (PP), RSMHV2 (P), and RSMHV2 (PP) were sequenced and compared with the known predicted MHV-A59 FP sequence (gene bank accession number: 9629812) and MHV2 (gene bank accession number: AF201929) FP within the spike gene fusion domain (Figure 1A). A consensus sequencing was carried out by RT-PCR of viral mRNA amplified from RSA59 (PP), RSMHV2 (P) and RSMHV2 (PP) infected L2 cells [17]. The FP sequence of RSA59 (PP) and RSMHV2 (P) was identical to the published sequence of our previous work [17]. RSMHV2 (PP) FP sequence was identical to RSMHV2 (P) with an addition proline in the FP (Figure 1B).

### 3.2. Divergent Fusogenicity of RSA59 (PP), RSMHV2 (P), and RSMHV2 (PP) in L2 Cells

Cell-to-cell fusion property (fusogenicity) of RSA59 (PP), RSMHV2 (P), and RSMHV2 (PP) was examined in the confluent culture of L2 cells at an MOI 0.5. RSA59 (PP), RSMHV2 (P), and RSMHV2 (PP) significantly differ in their fusogenic and cytopathic properties. Briefly, upon RSA59 (PP) infection, L2 cells started to fuse as early as 8 h and formed profuse syncytia which started to increase in size with time until 16 h post-infection (p.i.) (Figure 1C). Beyond16 h p.i., most of the giant syncytia started to dissolve as all of the infected cells had lysed. In contrast, RSMHV2 (P) infected individual cells rarely formed syncytia even 24 h after p.i., only a few infected cells (2–3) formed a cluster as observed previously (Figure 1D) [17]. Interestingly, infection of RSMHV2 (PP) with two consecutive prolines in the FP significantly increased the size of the multinucleated cluster of cells denoted as syncytia formation, as compared to RSMHV2 (P) (Figure 1E). The growth curve performed for RSMHV2 (P) and RSMHV2 (PP) until 24 h starting from the point of adherence (0 h) is identical hence depicting that no significant alteration in viral replication and growth can be observed after the addition of one proline (Appendix A). The experiments were repeated 5 times under the same conditions, and in most of the cases, distinctive multinucleated syncytia were observed to be formed by RSMHV2 (PP), which is different from RSMHV2 (P), but on the other hand, the number of multinucleated cells in the syncytia are much less compared to RSA59 (PP). The quantification of the fusion index as discussed in “Material and Methods” was plotted in a scatter-bar diagram (Figure 1F).

### 3.3. RSA59 (PP), RSMHV2 (P), and RSMHV2 (PP) Differ in Their Neuronal Tropism, Spread through Neuron and Syncytia Formation in Primary Neuronal Culture, Neuroblastoma Cell Line, Neuro2a, and Delayed Brain Tumor (DBT), Astrocytoma Cell Lines

Mixed neuroglial cultures enriched in primary neurons as discussed in “Material and Methods” were immunolabeled with anti-MAP2 antibody (neuron marker), anti-NFM antibody (neuron marker), anti-GFAP antibody (astrocyte marker), anti-H8H9 antibody (matured oligodendrocyte marker). Cells were counterstained with DAPI (nuclear stain). Intrinsic EGFP fluorescence denotes viral antigen. Visual manual counting of immunostained cells revealed that 65–70% of the cells in the culture are MAP2 positive indicating neurons (Figure 2D). Neurofilament neuronal staining frequently showed colocalization with MAP2 (Figure 2A). Immunostaining of the culture with astrocyte marker GFAP and differentiated oligodendrocyte marker H8H9 revealed that some cells are astrocytes and oligodendrocytes, respectively (Figure 2B,C). Visual manual counting of immunostained cells showed that 15–20% cells are GFAP positive astrocytes and 8–10% cells are H8H9 positive oligodendrocytes. Scatter-bar diagram in (Figure 2D), depicts the percentage of different neuroglial cells in the mixed neuroglial culture. These primary neuron enriched cultures were infected with RSA59 (PP), RSMHV2 (P), and RSMHV2 (PP) at an MOI of 2, and after 24 h p.i. cells were stained with neuronal marker MAP2 and counterstained with DAPI. Colocalization of intrinsic viral-EGFP fluorescence with neuronal markers revealed that a large number of MAP2+ neurons are infected, forming profuse syncytia. RSMHV2 (P) can infect neurons at a much lower efficiency. In contrast, the efficiency to infect neurons is significantly increased in RSMHV2 (PP). Moreover, colocalization studies revealed that RSA59 (PP) can infect neurons and also can spread through neurons to cause cell-to-cell fusion (Figure 2E–I). RSMHV2 (P) has limited ability to infect neurons and even if it infects the viral antigen spread is significantly restricted in the culture (Figure 2J–N). The infectivity and the viral spread increased in RSMHV2 (PP) compared to RSMHV2 (P) but remains much less for RSA59 (PP) as also observed in L2 cell infection (Figure 2O–S). Detailed quantification of differential neuronal tropism has been depicted in a scatter-bar diagram (Figure 2T). In summary, the results demonstrate that the addition of one proline significantly increases the neuronal tropism and viral antigen spread in RSMHV2 (PP) compared to its parental strain RSMHV2 (P), still, the addition of one proline in the backbone of RSMHV2 (P) was not able to match the ability to infect neuron, viral antigen spread and, cell-to-cell fusion as observed in RSA59 (PP).

The differential neuronal tropism of RSA59 (PP), RSMHV2 (P), and RSMHV2 (PP) were further confirmed in Neuro2a, neuroblastoma cells in continuous culture. Detailed immunofluorescence characterization revealed that 95% of Neuro2a cells in culture are NFM/MAP2 positive (data not shown), demonstrating homogeneity of neuronal cells in culture. Briefly, 85–90% of confluent Neuro2a cells were infected with RSA59 (PP), RSMHV2 (P), and RSMHV2 (PP) at 0.5 MOI. Time kinetic studies of cell-to-cell fusion from 12–24 h revealed that RSA59 (PP) profusely infected Neuro2a cells in culture and started to form syncytia as early as 12 h p.i. which increased with time to form giant syncytia as shown in Figure 3A–D until 24 h p.i. RSMHV2 (P) and RSMHV2 (PP) showed individually infected cells at 12 h but not many obvious multinucleated cells were observed in any of the infected cultures (Figure 3E,I). Whereas, after 16 h p.i. RSMHV2 (P) showed 2–4 cells in the cluster, no obvious syncytia were observed (Figure 3F,G). The rate of syncytia formation did not alter much even after 24 h p.i. (Figure 3H). In contrast RSMHV2 (PP) at 16 hp.i. formed multinucleated moderate size syncytia comprising 10–15 nuclei (Figure 3J,K). The size and number of the syncytia did not increase at 24 h p.i. in infected cells (Figure 3L). A scatter-bar diagram of the mean number of nuclei or cells per cluster clearly shows the significant difference in the size of the syncytia between RSA59 (PP) with RSMHV2 (P) or RSMHV2 (PP) at 12 h,16 h, and 24 h p.i. (Figure 3M). But the differences between RSMHV2 (P) and RSMHV2 (PP) were maximum and significant only at 16 h p.i. (Figure 3M). Time kinetics studies in Neuro2a further confirmed that RSA59 (PP) with RSMHV2 (P) and RSMHV2 (PP) differ in their neuronal tropism, viral spread, and syncytia formation. Further, these studies confirm that the addition of one proline in RSMHV2 (PP) significantly increases the efficiency of syncytia formation compared to parental strain RSMHV2 (P), but is significantly less able to form syncytia compared to RSA59 (PP).

Our observations in the neuron-enrich primary culture as well as neuronal cell line, Neuro2a culture where the addition of one proline in RSMHV2 (P) significantly increased syncytia formation though smaller in size and fewer in number than the syncytia observed in RSA59 (PP) prompted us to investigate the fusogenicity of RSMHV2 (PP) in astrocytoma Delayed Brain Tumor (DBT) cells. The DBT cells are known to be frequently used for the productive replication of different MHV strains. Confluent monolayers of DBT cells were infected with RSA59 (PP), RSMHV2 (P), or RSMHV2 (PP) at MOI 0.5. Similar to Neuro2a cells, RSA59 (PP) started forming giant syncytia as early as 8 h p.i. in the DBT cell line (Figure 4A), and that significantly increased in size and number by 12 h p.i. (Figure 4B), and right after that started to dissolve due to profuse viral replication. Like neuronal cell infection, RSMHV2 (P) and RSMHV2 (PP) can efficiently infect DBT cells in culture, which increases with time between 8 h–12 h p.i. At 8 h p.i. no multinucleated cell clustering was observed either in RSMHV2 (P) or RSMHV2 (PP) infected culture (Figure 4C,E). Clustering of multinucleated cells was significantly observed in the RSMHV2 (PP) cluster at 12 h p.i. (Figure 4F); whereas, RSMHV2 (P) infected cells still failed to form clusters (Figure 4D). The size and number of RSMHV2 (PP) clustering remained low as compared to RSA59 (PP) clustering. Quantification of the mean number of cells per syncytium at 8 h and 12 h p.i.is depicted in a scatter-bar diagram (Figure 4G).

### 3.4. RSMHV2 (P) and RSMHV2 (PP) Differ in Their Ability to Induce Necrotizing Hepatitis at Day 3 and Day 6 p.i.

Four-week-old C57BL/6J mice were infected intracranially with a 50% of LD50 dose of RSMHV2 (P) at 100 pfu and RSMHV2 (PP) at 20,000 pfu. Upon infection, on day 3 and day 6 p.i. mice were euthanized, and liver tissues were harvested for pathological studies. Briefly, harvested liver tissues from RSMHV2 (P), and RSMHV2 (PP) infected mice were processed, 5 µm thin longitudinal sections were stained with Hematoxylin and Eosin (H &E). Bright-field microscopic observations revealed that both RSMHV2 (P) and RSMHV2 (PP) induce large foci of hepatic lesions throughout the liver sections. Detailed quantification of the area of the hepatic lesions showed that the size of the hepatic lesions was much bigger in RSMHV2 (PP), at day 3 p.i. (Figure 5B), but the differences in the size of the hepatic lesions are much more evident at day 6 p.i. (Figure 5A) than RSMHV2 (P) (Figure 5A,C). The quantification data has been plotted in a scatter-bar diagram (Figure 4E). RSMHV2 (P), and RSMHV2 (PP) induce moderate to severe hepatitis like their parental stain but the insertion of one proline significantly increased the rate of hepatitis as denoted by the increase in the size of the hepatic lesion both at day 3 and day 6 p.i.

### 3.5. RSMHV2 (P), and RSMHV2 (PP) Differ in Their Ability to Distribute Viral Antigen in Different Neuroanatomic Regions on Day 3 and Day 6 p.i.

Four-week-old C57BL/6J mice were infected intracranially with a 50% of LD50 dose of RSMHV2 (P) at 100 pfu and RSMHV2 (PP) at 20,000 pfu. Day 3 and day 6 p.i. mice were euthanized, and brain tissues were harvested for pathological studies. Briefly, harvested brain tissues from RSMHV2 (P), and RSMHV2 (PP) infected mice were processed, 5–10 µm thin mid-sagittal sections were examined for the viral antigen distribution in different neuroanatomical regions. Brain sections were immunohistochemically stained with an anti-nucleocapsid antibody, Anti-N. Immunostaining data revealed that on day 3 RSMHV2 (P) infected brain samples viral antigen distribution is restricted mainly in the olfactory bulb, meninges, near the lateral-ventricle/ subventricular zone at the lining of the subependymal layer of the 4th ventricle (Figure 6A,C,E,G,I,K). As compared to RSMHV2 (P), on day 3 RSMHV2 (PP) infected brain section show more viral antigen that are invading the different neuroanatomical regions including the site of inoculation (lateral geniculate nuclei), olfactory bulb, meninges, and within the brain parenchyma, in the basal forebrain, ventral striatum, lateral ventricular nuclei and the subependymal layer of 4th ventricle (Figure 6B,D,F,H,J,L). Though no significant differences in the neuroanatomic spread of RSMHV2 (P) and RSMHV2 (PP) were observed, it seems that RSMHV2 (PP) invades the brain parenchyma with higher efficiency and replicates profusely in the olfactory bulb, basal forebrain, ventricular striatum, and cerebral cortex as compared to RSMHV2 (P).

On day 6 p.i.of mice, the diffused viral antigen is mainly present in the olfactory bulb and posterior region of the brain, brain stem in RSMHV2 (P) (Figure 6M,O,Q,S,U,W). In contrast, in RSMHV2 (PP) viral antigen was still present in the olfactory bulb, cerebral cortex, ventral striatum/basal forebrain, lateral ventricle, and brainstem region (Figure 6N,P,R,T,V,X). Much widespread dissemination of virus antigen in RSMHV2 (PP) infected mouse brain section compared with RSMHV2 (P) both at day 3 and day 6 p.i. indicated that the addition of one proline resulted in increased viral antigen spreading and proliferation. The viral antigen distribution of RSMHV2 (PP) and RSMHV2 (P) was compared in different neuroanatomical regions of the brain as mentioned in the “Material and Methods” section. RSMHV2 (PP) versus RSMHV2 (P) significantly differed in the amount of viral antigen in the olfactory bulb and cerebral cortex on day 3 p.i. On day 6 p.i. in addition to the olfactory bulb and cerebral cortex there are significant changes in the ventral striatum and brain stem region. No significant differences were observed at the lateral ventricle/subventricular zone on day 3 and day 6 p.i. The mean differences and standard error of the distribution of viral antigen in different anatomical regions are plotted as a scatter-bar diagram (Figure 6Y,Z).

### 3.6. RSMHV2 (P) and RSMHV2 (PP) Differ in Their Ability to Neuronal Spread from Gray Matter to White Matter and Invade into the White Matter in Spinal Cord

Four-week-old C57BL/6J mice were either mock-infected or infected intracranially with a 50% of LD50 dose of RSMHV2 (P) at 100 pfu and RSMHV2 (PP) at 20,000 pfu. Day 3 and day 6 p.i. mice were euthanized, and spinal cord tissues were harvested for histopathological studies. Briefly, harvested spinal cord tissues from Mock-infected, RSMHV2 (P), and RSMHV2 (PP) infected mice were processed, 5–10 µm thin cross-sections were examined for the viral antigen distribution in gray and white matter neurons (Figure 7A–F). Spinal cord cross-sections were immunostained with Anti-N indicating viral antigen. Mock infected spinal cord sections were negative for viral antigen staining (Figure 7A,D). The viral spread of RSMHV2 (P) as previously shown was restricted mainly to the gray matter and occasionally to the gray-white matter junction at day 3 and day 6 p.i.(Figure 7B,E). In contrast, in the RSMHV2 (PP) infected spinal cord viral antigen is present in the gray matter, gray-white matter junction, and also in the white matter of the spinal cord. RSMHV2 (PP) is significantly replicating and spreading to gray-white matter junctions and white matter as compared to RSMHV2 (P), but the differences are much more pronounced at day 6 p.i. than on day 3 p.i. (Figure 7C,F). The quantification data has been presented in a scatter-bar diagram (Figure 7G).

To confirm the viral antigen spread, spinal cord cryosections were prepared as described in the “Material and Methods” section. Intrinsic EGFP fluorescence was used to detect viral antigen spread in gray matter, and from gray to white matter. The neuroanatomical distribution of the gray-white matter was corroborated by its corresponding LFB stained sections which clearly demarcated gray matter from white matter in the spinal cord (Figure 7H). RSA59 (PP) is a known fusogenic demyelinating strain of MHV and is known to spread from gray matter to white matter by day 5 p.i. Previous studies demonstrated that the successful axonal transport of virus from gray matter to white matter p.i. is key to early-stage myelitis and chronic stage demyelination– given this reason it was used as a control for the rest of the studies [26,27]. In this study, we compared the viral antigen distribution of RSMHV2 (PP) and RSMHV2 (P) with RSA59 (PP) in cryosections. Data revealed that while in RSMHV2 (P) infected spinal cord sections at day 5 viral antigen was restricted to the gray matter (Figure 7J), in RSMHV2 (PP) viral antigen was also primarily present in the gray matter with occasional distribution from gray matter to white matter like RSA59 (PP) (Figure 7I). Insertion of proline in RSMHV2 (PP) increased the ability of RSMHV2 (PP) to traffic to the white matter but could not attain the ability of RSA59 (PP) to invade the white matter (Figure 7K).

### 3.7. RSMHV2 (PP) and RSMHV2 (P) Differ in Their Axonal Transport Contributing to Viral Antigen Spread from Gray Matter to White Matter

RSA59 (PP) is known to follow intraneuronal transport from gray to white matter and release at the nerve ending to infect white matter oligodendrocytes [26,27]. Whereas RSMHV2 (P) is known to infect neurons but is impaired in axonal transport and is not released at the nerve end. To examine whether the addition of one proline significantly altered the axonal transport of RSMHV2 (PP), spinal cord cryosection was immunostained with MAP2 and compared with RSA59 (PP) and RSMHV2 (P) infected spinal cord cryosections. Intrinsic EGFP fluorescence denotes viral antigen. Most of the colocalized images were taken from gray-white matter junctions to follow the axonal transport either from the dorsal or the ventral horn. Colocalization studies demonstrated that in RSA59 (PP) viral antigen colocalizes with MAP2 denoting the axonal translocation (Figure 8A–C,S,D,F,T). In RSMHV2 (P) not much colocalization was obvious either in the dorsal horn or the ventral horn (Figure 8G–L,U,V). Interestingly, RSMHV2 (PP) occasionally colocalized with MAP2 which demonstrates that RSMHV2 (PP) can conduit through the axon to reach the white matter (Figure 8M–R,W,X). Colocalization studies demonstrate that the addition of one proline significantly increased the ability of axonal transport of RSMHV2 (PP), and thus may allow it to invade the white matter.

### 3.8. RSMHV2 (PP) Induced Mild to Moderate Myelin Damage Compared to No Demyelination of RSMHV2 (P), but the Intensity of the Demyelination Was Much Less Compared to Parental Demyelinating Strain RSA59 (PP) as Studied on Day 30 p.i.

Four-week-old C57BL/6J mice were either mock-infected or infected intracranially with 50% of LD50 dose of RSA59 (PP) at 20,000 pfu, RSMHV2 (P) at 100 pfu, and RSMHV2 (PP) at 20,000 pfu. At the chronic stage, day 30 p.i. mice were euthanized and spinal cord tissues were harvested for histopathological studies. Briefly, harvested spinal cord tissues from mock-infected, RSA59 (PP), RSMHV2 (P), and RSMHV2 (PP) infected mice were processed, 5–10 µm thin cross-sections were stained with Luxol Fast Blue (LFB) to examine for demyelination. No evident demyelination was observed in mock-infected spinal cord sections (Figure 9A). RSA59 (PP), as known in previous studies produces discrete large demyelinating plaques to profuse confluent demyelination both in the dorsal column, the anterior and ventral horn of white matter at multiple levels of spinal cord tissue [26,30]. Representative images are shown in Figure 9B. No obvious myelin loss was observed in RSMHV2 (P) infected spinal cord sections (Figure 9C), in contrast, RSMHV2 (PP) infection-induced sparse myelin loss in dorsal column, ventral/anterior horn but no large discrete plaques or confluent plaques were observed throughout the spinal cord section (Figure 9D). The quantification of demyelination was performed as discussed in the “Material and Methods” section and the scatter-bar diagram of the quantified data demonstrates that RSMHV2 (PP) induced significant demyelination compared to RSMHV2 (P), but the intensity of the myelin loss is significantly low compared to RSA59 (PP) (Figure 9E). The addition of one proline significantly increased the demyelinating properties of RSMHV2 (PP) compared to no demyelination by one proline containing parental strain RSMHV2 (P), but not sufficient enough to form demyelinating plaques observed in the case of RSA59 (PP).

### 3.9. Rigidity in the Fusion Peptide from the Addition of Proline to Spike Protein from RSMHV2 (P)

Molecular dynamics was used to the structural properties of the spike protein variants. The free energy landscape of spike from RSMHV2 (PP) has a single potential well like the spike from RSMHV2 (P), albeit shallower and in contrast to RSA59 (PP) spike which has two deep potential wells (Figure 10A). The torsional flexibility shows that the local dynamics of the FP from spike in RSMHV2 (PP) are almost identical to the variant from RSA59 (PP) (Figure 10B), also corroborated by the root-mean-square fluctuation. The secondary structure of the FP in RSMHV2 (PP) is also similar to the helix-loop-helix-loop helix arrangement in spike from RSA59 (PP). In contrast, the spike from RSMHV2 (P) is missing the central helix and has a helix-loop-helix arrangement (Figure 10B). The disorder is caused by the lack of H-bonds in the region due to the absence of double proline in the FP of spike from RSMHV2 (P). Despite the local flexibility of the FP in spike from RSMHV2 (P), the deep free energy well indicates a relatively rigid structure. This is explained by the fact that the free energy landscapes in Figure 10A encompass both the tertiary and quaternary dynamics of the spike protein fusion domain. Therefore, the local tertiary dynamics of spike from RSMHV2 (PP) resemble the spike from RSA59 (PP), but the global quaternary dynamics resemble spike from RSMHV2 (P).

## 4. Discussion

The presence of two consecutive central prolines (PP) within the FP contributes to the efficiency of the fusion process. The data from the current study clearly emphasizes this, given a non-fusogenic non-demyelinating strain RSMHV2 (P) is mutated to a fusogenic and weakly demyelinating strain RSMHV2 (PP) on insertion of an additional proline adjacent to the central proline in the FP. The extent of this difference can be gauged from the use of a 20,000 pfu inoculation dose for experimentation with RSA59 (PP) and RSMHV2 (PP), compared to only 100 pfu for RSMHV2 (P) in the in vivo study (Appendix A). This difference in inoculation volume is based on the feasibility of experimental design else the readout of syncytia formation becomes erroneous when a large pfu of the virus is used without commensurate virus fusion ability. In vitro comparison of RSMHV2 (PP) with its parental strain RSMHV2 (P) and RSA59 (PP) show different rates and intensity of infection with primary neonatal neuronal cells, Neuro2A cells, L2 cells, and Delayed Brain Tumor astrocytoma cells; but the neuroglial cell infection was significantly different. Overall, the RSMHV2 (PP) showed moderate but significant cell-to-cell fusion and syncytia formation compared to RSMHV2 (P), but significantly low fusogenicity compared to RSA59 (PP). In vivo studies also show consistent results, with RSMHV2 (PP) producing a severe necrotic lesion in the liver compared to the RSMHV2 (P)—the caveat being the 50% of the LD50 dose of 100 pfu used for RSMHV2 (P), compared to 20,000 pfu for RSMHV2 (PP) and RSA59 (PP). The insertion of an adjacent proline appears to boost 50% of the LD50 dose from 100 pfu to 20,000 pfu in mice.

The decreased susceptibility of RSMHV2 (PP) compared to RSMHV2 (P) may also occur considering that the cell-to-cell spread helped the mutant strain to evade the immune responses. Though this hypothesis is beyond the scope of this study, it would be interesting to understand whether the addition of one proline significantly altered the immune modulation in mounting the host immunity. Our studies demonstrate that though both RSMHV2 (PP) and RSMHV2 (P) differ in their susceptibility, they successfully invade brain parenchyma but significantly differ in viral antigen distribution in some of the neuroanatomic regions, namely, ventral striatum, basal forebrain, cerebral cortex, and brain stem. Some significant differences in the viral antigen distribution are observed in the olfactory bulb during day 3 p.i., but none on day 6 p.i. None are observed in the leptomeninges either, nor in the ventricular and subventricular regions of the brain indicating that RSMHV2 (PP) and RSMHV2 (P) do not differ in their ability to cause meningitis. While striking differences were observed in the viral antigen distribution of both the strains in the spinal cord, in RSMHV2 (P), the viral antigen is mainly restricted to the gray matter both at day 3 and 5 p.i, but RSMHV2 (PP) did not restrict to the gray-white matter demarcation and was almost equally distributed throughout the spinal cord. As RSMHV2 (P) was restricted to gray matter in the acute stages of infection, there was no myelin loss in the chronic phase. RSMHV2 (PP) showed moderate demyelination pertaining to the spread of the virus along the axon through the white matter in the acute stage of infection. The intensity of the demyelination of RSMHV2 (PP) is significantly less compared to the parental demyelinating strain RSA59 (PP).

The comparative role of PP in the aggressive virulence of RSA59 (PP) was previously investigated by deleting one proline in the PP of the FP, yielding strain RSA59 (P) [17]. RSA59 (P) significantly lowered neural cell syncytia formation and viral titers post-infection in vitro. Transcranial inoculation of C57Bl/6 mice with RSA59 (PP) or RSA59 (P) yielded similar degrees of necrotizing hepatitis and meningitis, but only RSA59 (PP) can produce widespread encephalitis extending deeply into the brain parenchyma. Both the virus variants are mostly cleared from the brain by day 6 post-infection. In addition, RSA59 (PP) is also known to cause optic neuritis and demyelination in the optic nerve as well as retinal ganglionic cells (RGC) loss. As a mechanism, it was evident that upon intracranial inoculation into the lateral geniculate nuclei of the brain, RSA59 (PP) in addition to anterograde axonal transport also follow retrograde axonal transport through the optic nerve to reach the retinal ganglionic cells and in due course causes optic neuritis, optic nerve demyelination, and RGC loss. Thus, an intact dyad of proline residues in the FP of the targeted-recombinant meningio-encephalomyelitis and demyelinating MHV strain may regulate the translocation of the virus-antigen along the axons and subsequent neurodegeneration.

Which FP feature is likely critical in establishing the virus phenotype—the location of the proline residue in the FP, or the physicochemical property of the amino acid at that position? Previous studies on MHV-A59 FP have shown that the substitution of the proline residue at position 938 with lysine (P938K) partially impaired fusion while replacing the same proline residue with a leucine residue did not have any effect on fusion [31]. The mutagenesis/substitution of the methionine residue at position 936 with lysine (M936K) or leucine (M936L) did not affect fusion [32]. Thus, the data suggest that both the location and the physicochemical property of the amino acid are important. The importance of location can be hypothesized from a recently proposed membrane-protein contact initiation model in SARS-CoV-2 Spike [33]. The FP in the Spike trimeric quaternary structure is juxtaposed in a location that facilitates its early contact with the host membrane after attachment with the host cell receptor. The trimeric architecture of Spike ensures three copies of the FP are present and the alignment of each of the FPs is midway between a pair of receptor binding domains that can guide the initial contact with the host membrane. The precise geometric/positional requirement of the proline appears to coincide with the positional requirement of FP for facile early membrane-protein contact. The presence of proline at the required position also highlights the importance of its physicochemical property. Proline is distinct due to its imino acid stereochemistry compared to an amino acid. Consequently, it cannot donate hydrogen bonds in the tertiary structure, and its side-chain ring is fused to the protein backbone thereby restricting the polypeptide chain flexibility around its location. Among all protein residues, proline restricts the protein backbone flexibility the most which can be seen from its smallest area occupancy in the Ramachandran map [34].

The presence of PP is expected to sterically create a larger segment of local backbone restriction compared to a single-proline in the FP. It is expected to induce rigidity also in its immediate spatial neighborhood. This is corroborated by the MD simulation of MHV2 spike protein where the PP in the FP region is significantly restricted in RSMHV2 (PP) compared to RSMHV2 (P). The local fluctuation graph around FP matches closely with RSA59 (PP). The local rigidity is also supplanted by more local hydrogen bonds in PP containing FP and is driven by the stereochemically-imposed rigidity as the single P containing FP in RSMHV (P) has limited hydrogen bonds. The local rigidity in the PP neighborhood is, however, quite contrasting to the global flexibility of the S2 domain. The S2 domain of RSMHV2 (PP) appears to have a shallower energy well compared to RSMHV2 (P), although both have a single potential well. This differs from the two energy wells from the S2 domain of RSA59 (PP). A greater number of wells in the energy landscape indicates more interconvertible conformational states of the protein and also reflects a low transition barrier when the lowest energy states in these wells are conformationally close to each other. This is favorable for triggering conformational transition needed for change to post-fusion structure in comparison to a deep single energy well as in RSMHV2 (P) where the chances of these transitions are dampened. This may explain the order of fusogenic potential seen in the three viruses considered in this study.

The critical location of the FP in the spike protein and its demonstrated ability to alter fusogenicity and pathogenesis make it an apt site for a potential antiviral. A series of studies exist that attempt to identify the minimum essential region required for the fusogenic property for designing the therapeutic mimetic peptides [34,35]. These premised that the internal FP owing to its hydrophobicity and location adjacent to the heptad repeat segments within the S2 spike domain of MHV-A59 may be responsible for the fusogenic properties associated with its hepato-neuropathogenicity [16,17,31,32]. Dissecting the minimal essential motif of the Spike protein FP may enable us to design a mimetic peptide to set the stage for competition to reduce virus-induced neuroinflammation. This strategy is advantageous because the S2 domain of spike is much more evolutionarily conserved compared to the receptor binding or the N-terminal domain. As such, this will also allow us to build a therapeutic approach that applies to pan-CoV situations.

Altogether, combining proline insertion deletion studies highlight insights that fusion-dependent viral spread could be important in general for virus-host cell interaction. Interestingly, many fusion proteins do not have a central proline in their FP; however, residues therein that improve FP rigidity and cause the neighborhood structure to become more ordered in the membrane environment improve kinetic efficiency of the fusion process, making them suitable targets as per insights of our study. Overall, our work is useful in understanding the mechanisms employed by the virus fusion apparatus and may help in guiding the development of therapeutic interventions to restrict cell-to-cell viral antigen spread and its consecutive viral infection.

## 5. Conclusions

The presence of two consecutive central prolines (PP) within the FP of spike protein of β-coronavirus contributes to the efficiency of the fusion process. Comparative in vitro and in vivo studies between virus strains RSA59(PP), RSMHV2 (P), and RSMHV2 (PP) in the FP demonstrate that the insertion of one proline significantly resulted in enhancing the virus fusogenicity, spread, and consecutive neuropathogenesis. The intensity of the demyelination as a read-out of the neuropathogenesis of RSMHV2 (PP) is significantly less compared to the parental demyelinating strain RSA59 (PP), but significantly more compared to RSMHV2 (P). The critical location of two central prolines of the FP predominantly determines fusogenicity and may be essential but not sufficient to cause demyelination.

## Figures and Tables

**Figure 1 viruses-14-00834-f001:**
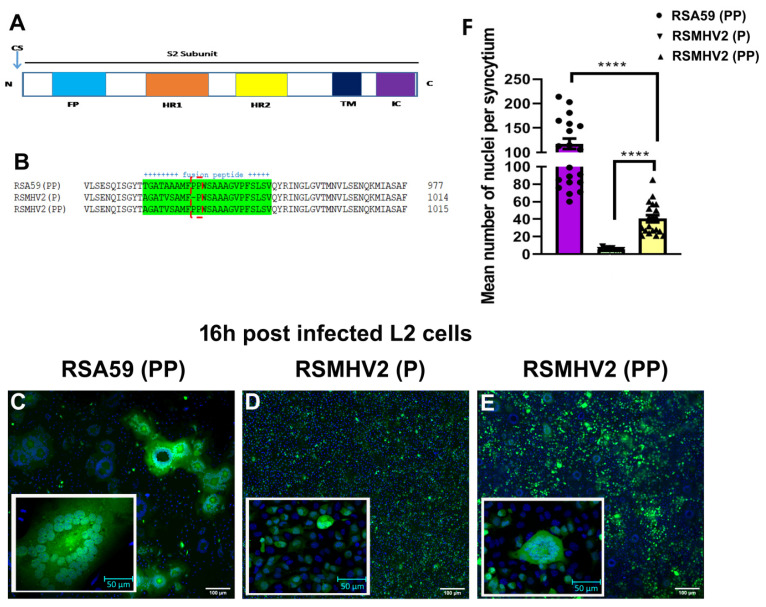
**Sequence comparison of fusion peptide and fusogenic properties of RSA59 (PP), RSMHV2 (P), and RSMHV2 (PP) in L2 cells**. Schematic of the S2 subunit of coronavirus Spike gene, showing cleavage site (CS), FP, heptad repeat 1 (HR1), heptad repeat 2 (HR2), transmembrane domain (TM), and intracellular tail (IC) (**A**). FP region of RSA59 (PP), RSMHV2 (P), and RSMHV2 (PP) sequences are aligned using ClustalW tool, dashed lines show central double proline (PP) or single proline (P) in RSA59 (PP), RSMHV2 (PP), and RSMHV2 (P), respectively (**B**). A monolayer of L2 cells was infected with RSA59 (PP), RSMHV2 (P), and RSMHV2 (PP) at MOI 0.5. The cells were incubated for 16 h at 37 °C with 5% CO_2_, fixed with 4% paraformaldehyde, and mounted in DAPI containing mounting media. Epifluorescence microscopy was performed, and images were acquired and further processed using Image J (Fiji) software. The green fluorescence is the intrinsic viral EGFP that is incorporated into the genome. Images of DAPI (blue) and EGFP (green) were merged to construct the images presented. RSA59 (PP) and RSMHV2 (PP) infection cause syncytia formation (**C**,**E**) but RSMHV2 (P) infection rarely formed syncytia (**D**). The mean nuclei per syncytium were quantified and scatter-bar plotted (**F**). Experiments were repeated five times. Significance level was taken at *p* < 0.05 following unpaired *t*-test analysis. **** *p* < 0.00001.

**Figure 2 viruses-14-00834-f002:**
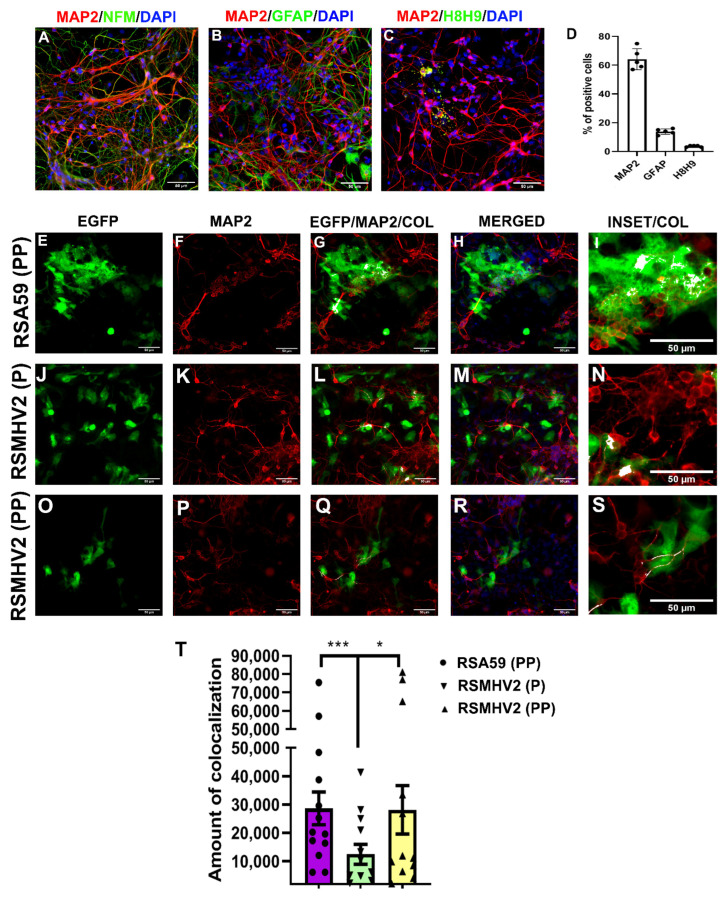
**Characterization and infection of primary neuronal culture with RSA59 (PP), RSMHV2 (P), and the one proline added mutant RSMHV2 (PP)**. Mixed neuroglial culture enriched in primary neuron immunolabeled with anti-MAP2 antibody (Red; neuronal marker), anti-NFM antibody (Green; neuron marker) (**A**), anti-GFAP antibody (Green; astrocytic marker) (**B**), anti-H8H9 antibody (Green; matured oligodendrocyte marker) (**C**). Cells were counterstained with DAPI (Blue; nuclear stain). (**D**), Visual manual counting of immunostained cells revealed that culture consists of 65% MAP2 positive neurons. Primary neuronal cultures were infected with RSA59 (PP) (**E**–**I**), RSMHV2 (P) (**J**–**N**), and RSMHV2 (PP) (**O**–**S**) at an MOI of 2, and after 24 h p.i. cells were stained with MAP2 and counterstained with DAPI. Representative images of the cells infected with the virus are shown where the EGFP panel shows the virus-infected cells, MAP2 channel denotes neurons in the culture and colocalization shows that the neurons were infected with the virus. The merged panel depicted the neurons are EGFP positive hence infected with the virus and DAPI represents the nucleus. Higher magnification images of the colocalization are highlighted in insets. (**T**) Colocalization of MAP2 and EGFP was represented in a scatter-bar plot. Experiments were repeated two times (N = 2) with three replicates per experiment per virus (n = 6). Significance level was taken at *p* < 0.05 following unpaired *t*-tests analysis. * *p* < 0.05, *** *p* < 0.001.

**Figure 3 viruses-14-00834-f003:**
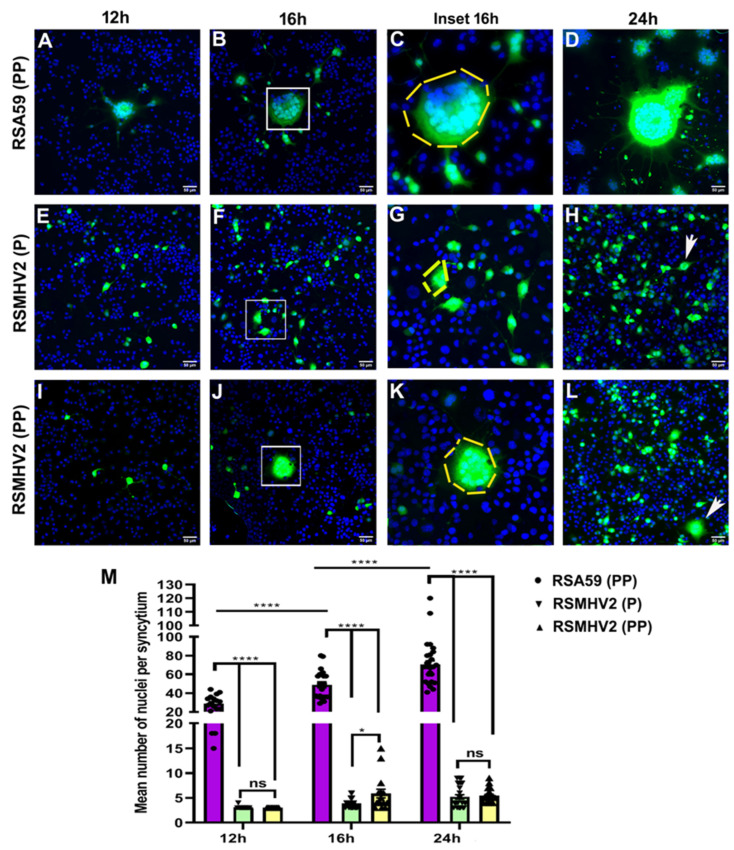
**Comparison of syncytia formation in Neuro2a cell infected by RSA59 (PP), RSMHV2 (P), and the one proline added mutant RSMHV2 (PP).** Neuro2a cells were infected with RSA59 (PP), RSMHV2 (P), and RSMHV2 (PP) at 0.5 MOI. The green fluorescence is due to EGFP that is integrated into the viral genome. Images of DAPI (Blue) and EGFP (Green) were merged to construct the final images presented here; (**A**–**D**) panels are from RSA59 (PP) infected Neuro2a cells at 12 h (**A**) 16 h, (**B**,**C**) is a magnified image of 16 h, (**D**) at 24 h, respectively. Panel E-H shows Neuro2a cells infected by RSMHV2 (P), (**E**) at 12 h, (**F**) 16 h, (**G**), is a magnified image of 16 h, (**H**) at 24 h, respectively. Arrowheads indicate 1–2 cells in a cluster. (**I**–**L**) show Neuro2a cells infected by RSMHV2 (PP) where (**I**) is 12 h p.i., (**J**) at 16 h, (**K**) magnified 16 h, and (**L**) 24 h p.i, respectively. The mean nuclei per syncytia were counted and plotted in a scatter bar (**M**). Experiments were repeated three times (N = 3) with three replicates per experiment per virus per time point p.i (n = 9). Significance level was taken at p < 0.05 following unpaired *t*-test analysis. * *p* < 0.05, **** *p* < 0.0001; ns means not significant.

**Figure 4 viruses-14-00834-f004:**
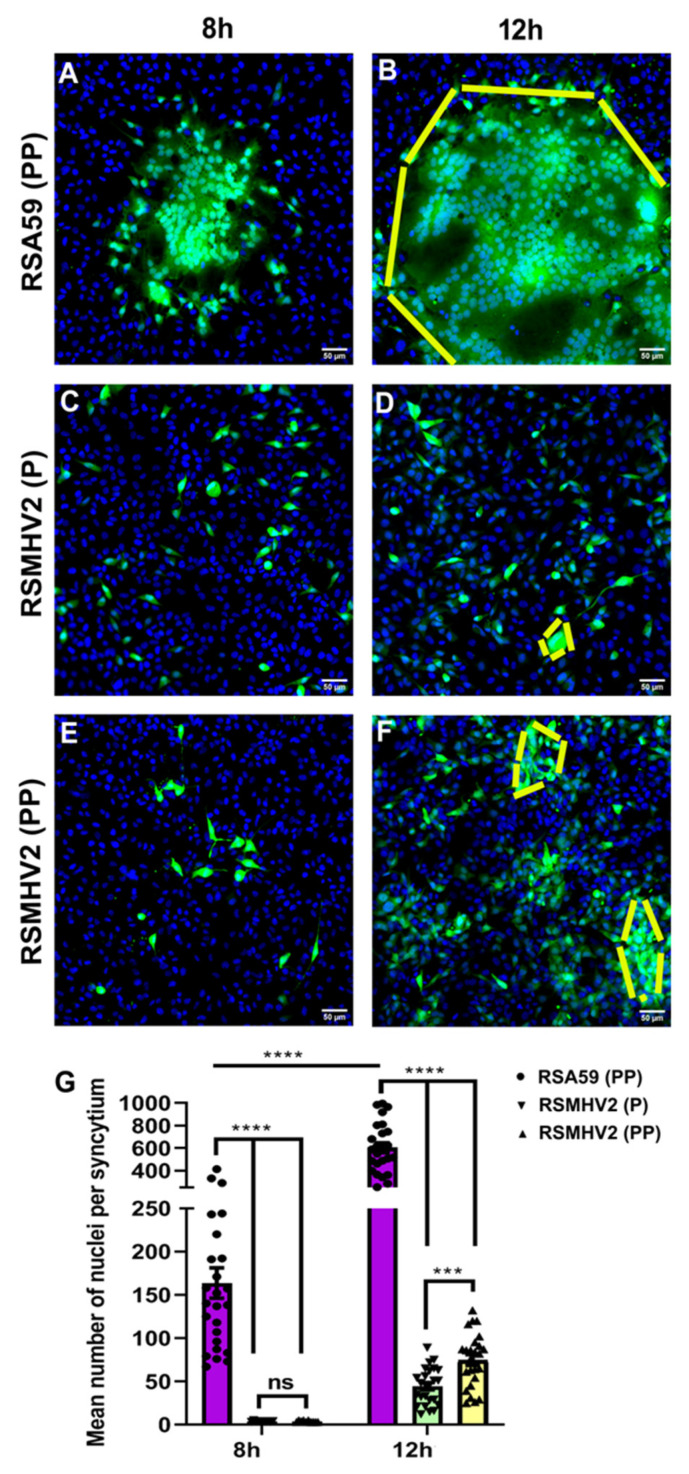
**Effect of double proline on the fusogenic ability of RSA59 (PP), RSMHV2 (P), and RSMHV2 (PP) in DBT cell line**. DBT cells were infected with RSA59 (PP), RSMHV2 (P), and RSMHV2 (PP) at an MOI of 0.5. Viral Antigen is denoted by EGFP fluorescence. Images are composite of DAPI and FITC channels; Panel (**A**,**B**) are from RSA59 (PP) infected DBT cells at 8 h and 12 h p.i, respectively. Large syncytia are indicated within the dashed lines. Panel (**C**,**D**) and (**E**,**F**) depicts DBT cells infected by RSMHV2 (P) and RSMHV2 (PP), respectively at 8 h and 12 h p.i. Dashed lines indicate clusters of infected cells in panels (**D**,**F**). The mean nuclei per syncytium in panels (**A**,**B**), and clusters of infected cells in panels (**C**–**F**) were counted and plotted in the scatter-bar diagram (**G**). Experiments were repeated three times (N = 3) with three replicates per experiment per virus per time point p.i (n = 9). Significance level was taken at *p* < 0.05 following unpaired *t*-tests analysis. *** *p* < 0.001; **** *p* < 0.0001.

**Figure 5 viruses-14-00834-f005:**
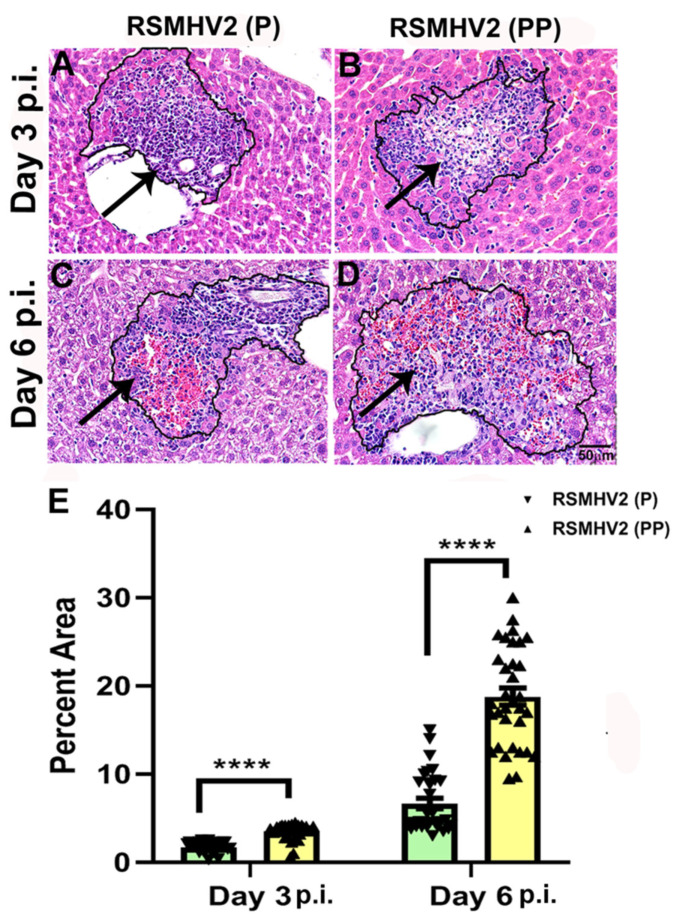
**Liver pathology consisting of moderate to severe hepatitis following RSMHV2 (P) and RSMHV2 (PP) infection**. Five-micron thin liver sections of day 3 and day 6 post-infected mice were stained with Hematoxylin & Eosin. Liver pathology consists of moderate to severe necrotizing and non-necrotizing hepatitis in both RSMHV2 (P) (**A**,**C**) and RSMHV2 (PP) (**B**,**D**) at day 3 and day 6 p.i., respectively. The size of hepatic lesions was quantified on day 3 and day 6 p.i., and the percent area of hepatic lesions was plotted in the scatter-bar diagram (**E**). The level of significance was calculated by unpaired Student’s *t*-test with Welch’s correction. n = 6 mice each day per virus infection and **** *p* < 0.0001.

**Figure 6 viruses-14-00834-f006:**
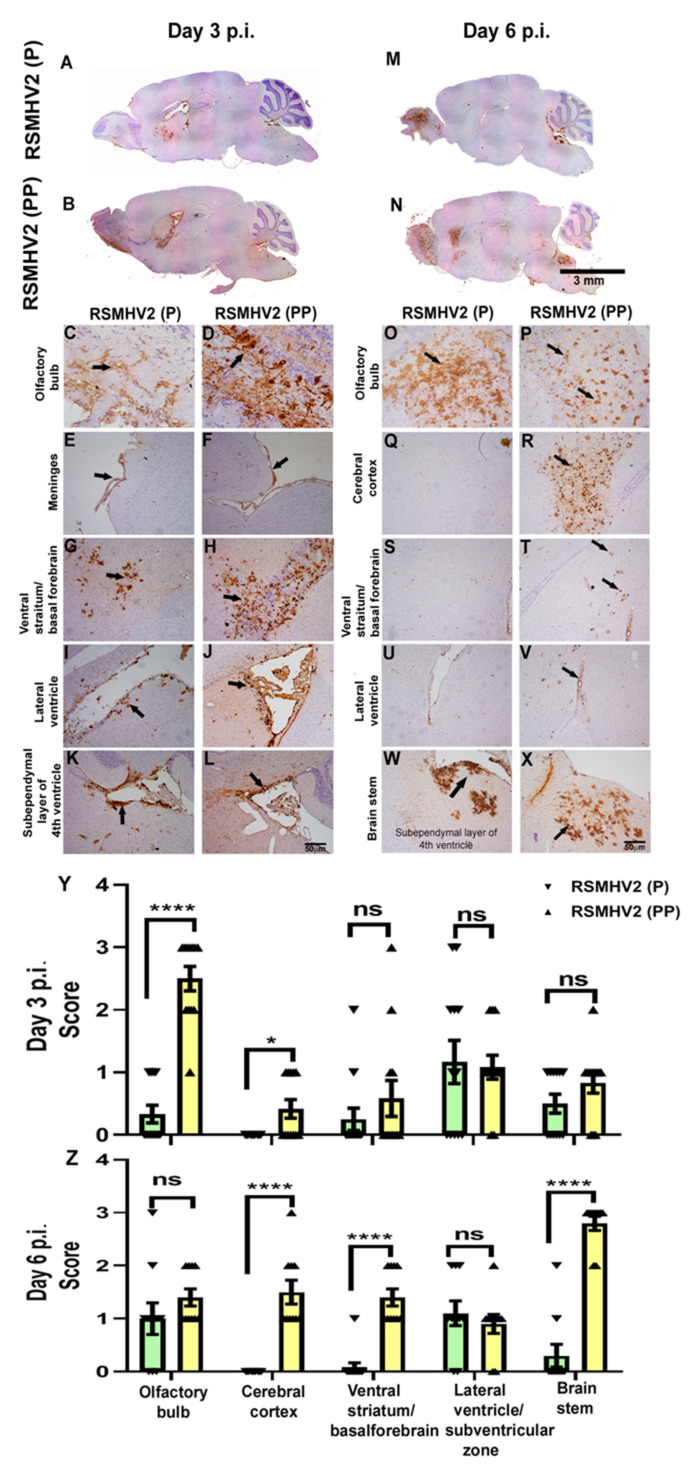
**Differential distribution of viral antigen in different neuroanatomical regions from RSMHV2 (P) and RSMHV2 (PP) at days 3 and 6 p.i. brains**. Five-micron thick mid-sagittal brain sections from RSMHV2 (P) and RSMHV2 (PP) infected mice were immunohistochemically stained with anti-N antibody (viral antigen). Scanned images of a whole mid-sagittal section of the brain from RSMHV2 (P) (**A**) and RSMHV2 (PP) (**B**) on day 3 and (**M**,**N**) at day 6, respectively. On day 3, both RSMHV2 (P) and RSMHV2 (PP) infect and spread to similar neuroanatomic regions but the amount of viral antigen was significantly more in RSMHV2 (PP) compared to RSMHV2 (P). In RSMHV2 (P), viral antigen distribution was observed in the olfactory bulb (**C**), the subpial layer of meninges (**E**), ventral striatum/basal forebrain (**G**), lateral ventricle (**I**), and the subependymal layer of the 4th ventricle (**K**). Similarly, in RSMHV2 (PP) infected brain sections, viral antigen distribution was observed in different regions: olfactory bulb (**D**), meninges (**F**), ventral striatum/basal forebrain (**H**), lateral ventricle (**J**), and also to the subependymal layer of 4th ventricle (**L**). On day 6 p.i, both RSMHV2 (P) and RSMHV2 (PP), viral antigen distribution was observed in the olfactory bulb (**O**,**P**), and the subependymal layer of the 4th ventricle (**W**,**X**). Meanwhile, RSMHV2 (PP) was able to invade deep in parenchyma compared to RSMHV2 (P) in the cerebral cortex (**Q**,**R**), ventral striatum/basal forebrain (**S**,**T**), and lateral ventricle (**U**,**V**), and brainstem. Arrows indicate viral antigen staining. Quantification was done based on the scores 0: no infection; 1: very small foci of infection; 2: widespread small foci of infection; 3: widespread with large foci of infection. Day 3 p.i (**Y**) showed a significant difference in the olfactory bulb and cerebral cortex. Day 6 p.i. (**Z**) revealed a significant difference in the cerebral cortex, ventral striatum/basal forebrain, and brainstem (N = 6 mice per virus infection).The level of significance was determined by unpaired Student’s *t*-test with Welch’s correction * *p* < 0.05, **** *p* < 0.0001.

**Figure 7 viruses-14-00834-f007:**
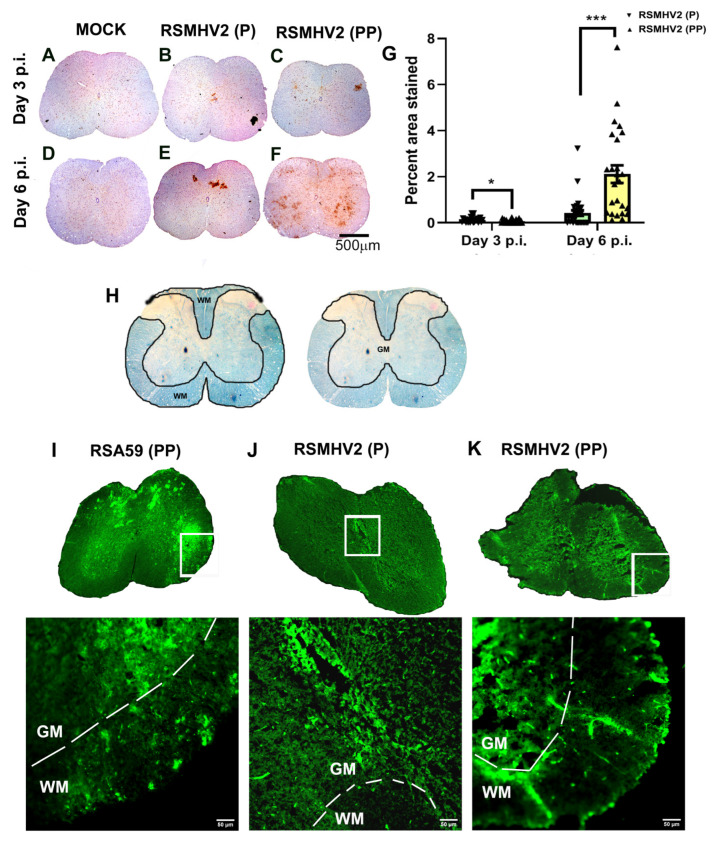
**Differential distribution of viral antigen in RSMHV2 (P) and RSMHV2 (PP) infected mice spinal cord**. Day 3 and Day 6 p.i. mock and viral infected spinal cord cross-sections were stained immunohistochemically (IHC) with anti-N antibody (**A**–**F**). No obvious staining was observed in any mock-infected sections (**A**,**D**). On day 3 p.i. in RSMHV2 (P) infected spinal cord viral antigen distribution was observed restricted around the central canal (**B**), whereas in RSMHV2 (PP) viral antigen distribution was apparent throughout the spinal cord including gray matter, gray-white matter junction with occasional distribution to the white matter (**C**). On day 6 p.i. RSMHV2 (PP) replicated profusely in the gray matter also with frequent distribution to the white matter of spinal cord (**F**) but in RSMHV2 (P) infected spinal cord, viral antigen was restricted mostly in the gray matter (**E**). Cross-section of spinal cord showing white and gray matters mapped out in an LFB staining (**H**). Day 5 p.i. infected spinal cord cryosections were prepared for fluorescent microscopy. Dashed lines demarcate the grey and white matter in cryosections denoted by GM and WM, respectively. RSA59 (PP) infected spinal cord showed widespread viral antigen indicated by EGFP in both grey and white matter (**I**). Similar to IHC, RSMHV2 (PP) infected spinal cord showed viral antigen present in gray and white matter (**K**) whereas RSMHV2 (P) infected spinal cord showed a restricted spread of viral antigen around the central canal in gray matter with no viral antigen spread to the white matter (**J**). The viral antigen staining was quantified and plotted as a scatter-bar diagram (**G**). The level of significance was determined by unpaired Student’s *t*-test with Welch’s correction. N = 6 mice per day per virus infection. * *p* < 0.05; *** *p* < 0.001.

**Figure 8 viruses-14-00834-f008:**
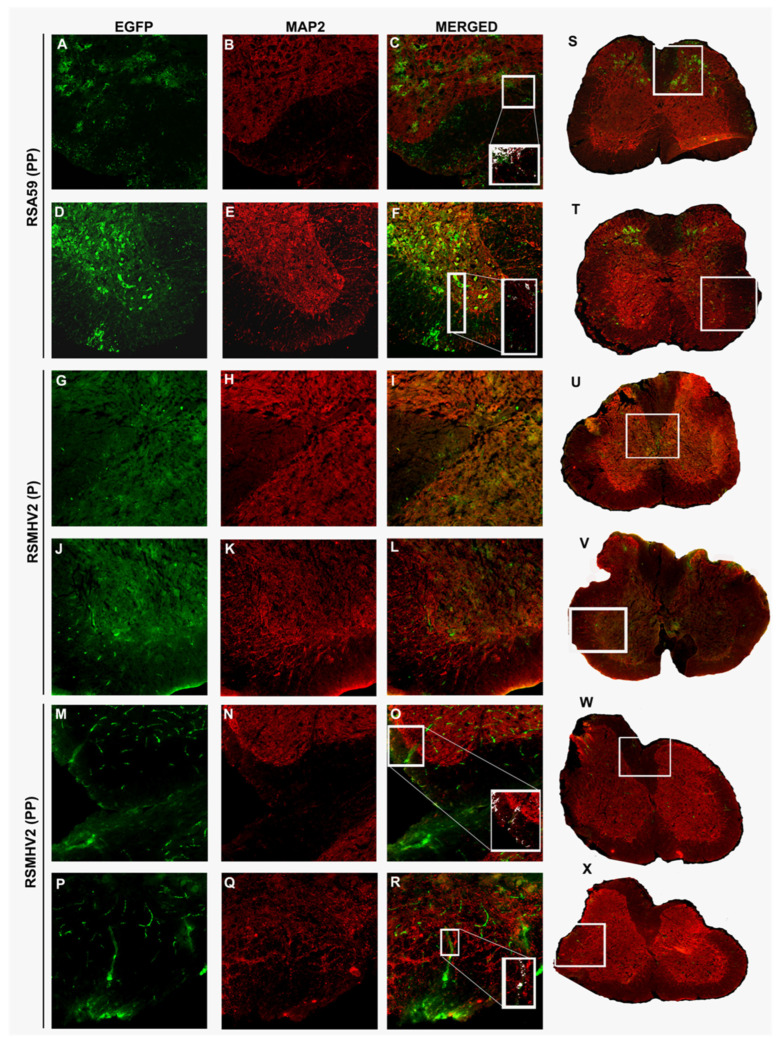
**Colocalization of viral antigen with neuronal microtubule marker (MAP2) of RSA59 (PP), RSMHV2 (P), and RSMHV2 (PP) infected spinal cord cryosections**. Day 5 post-infected spinal cord cryosections immuno-fluorescently labeled with MAP2 (Red) (**B**,**E**,**H**,**K**,**N**,**Q**). Intrinsic EGFP fluorescence denotes viral antigen (**A**,**D**,**G**,**J**,**M**,**P**). Merged images of EGFP and MAP2 (**C**,**F**,**I**,**L**,**O**,**R**). RSA59 (PP) infected spinal cord sections are (**A**–**F**,**S**,**T**), RSMHV2 (P) are from (**G**–**L**,**U**,**V**), RSMHV2 (PP) are (**M**–**R**,**W**,**X**). Dorsal column gray-white matter junctions are in panels (**A**–**C**) (RSA59 (PP)), (**G**–**I**) (RSMHV2 (P)), and P-R (RSMHV2 (PP)). The ventral white matter and adjacent ventral horns from the infected section are shown as RSA59 (PP) (**D**,**F**), RSMHV2 (P) (**J**–**L**), RSMHV2 (PP) (**P**–**R**). The insets in the merged images show magnified views of the co-localization of virus-infected cells with the Anti-MAP2 and dendrites at the gray-white matter junction. In the insets, co-localization is seen as white with infected axons.

**Figure 9 viruses-14-00834-f009:**
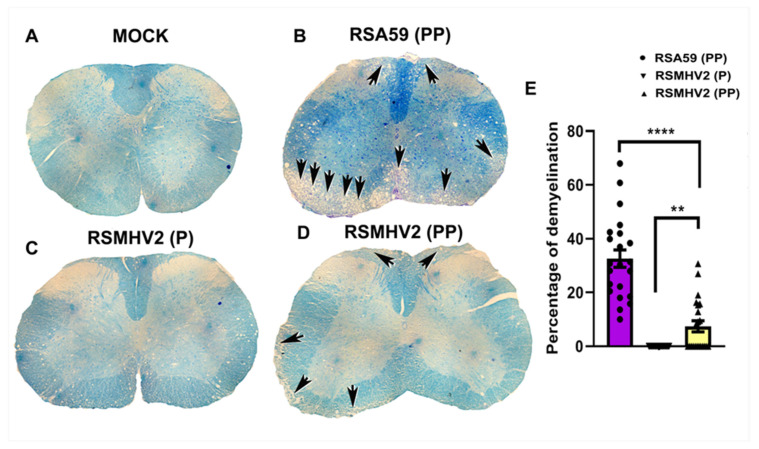
**Presence of double proline in the FP of RSMHV2 (PP) induces sparse myelin loss during day 30 p.i. (chronic stage of infection)**. Five to ten-micron thick spinal cord sections from either mock-infected (**A**) or RSA59 (PP) (**B**), RSMHV2 (P) (**C**), RSMHV2 (PP) (**D**) infected mice were stained with LFB to detect myelin loss. No myelin loss was observed in mock-infected or RSMHV2 (P) infected spinal cord sections (**A**,**C**), respectively. In contrast, large demyelinating plaques were observed at multiple levels of the spinal cord from RSA59 (PP) infected mice (**B**). RSMHV2 (PP) infected mice showed discrete yet fewer demyelinating axons without any obvious demyelinating plaques in sections of the spinal cord (**D**) (Black arrows indicate demyelinated area). Quantification of demyelination was plotted in the scatter-bar diagram (**E**), and the level of significance was calculated by unpaired Student’s *t*-test. N = 5 mice per virus infection. ** *p* < 0.001; **** *p* < 0.0001.

**Figure 10 viruses-14-00834-f010:**
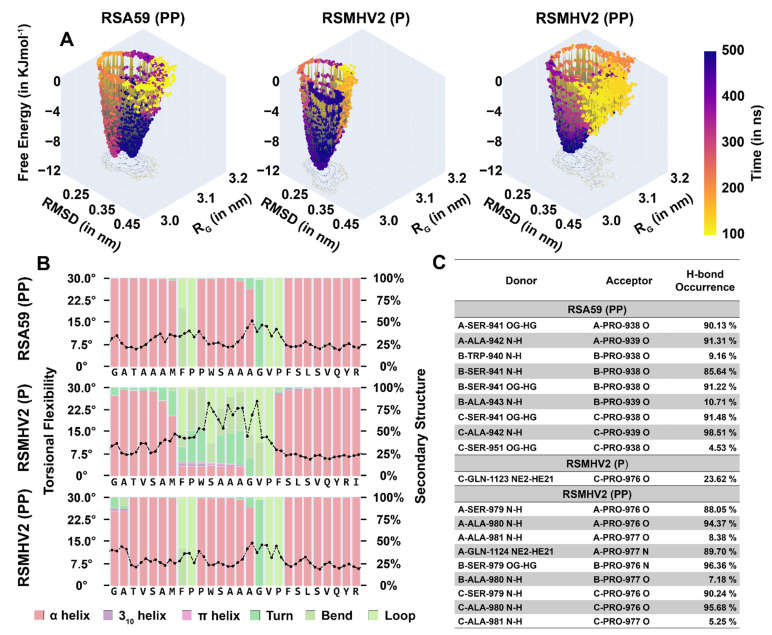
**Results from the analysis of MD simulations of Spike from RSA59 (PP), RSMHV2 (P), and RSMHV2 (PP) in water**. (**A**) Free energy landscapes of spike proteins from RSA59 (PP), RSMHV2 (P), and RSMHV2 (PP) from root-mean-square deviation and radius of gyration of the trajectory between 100–500 ns. The trajectory points are plotted on the surface and colored according to time as given by the color bar. (**B**) Torsional flexibility and secondary structure of the fusion domain in the three simulations. The two markers for each residue correspond to φ and ψ backbone torsion angle. The stacked bar for each residue shows the percentage of frames for which the secondary structure was observed. Since the three chains of the spike protein are identical and their results were similar, the results for the three chains were averaged. (**C**) Hydrogen bonds with the central proline/prolines in the FP. The donor and acceptors are given in the following format ‘Chain-Residue-Residue number Atom name’; the donor is also suffixed with ‘-Hydrogen atom name’.

## Data Availability

Not applicable.

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
