# Peer review of "Two Consecutive Prolines in the Fusion Peptide of Murine β-Coronavirus Spike Protein Predominantly Determine Fusogenicity and May Be Essential but Not Sufficient to Cause Demyelination"

_viruses, 2022, doi:10.3390/v14040834_

Round 1

Reviewer 1 Report

The study investigated into the molecular mechanism of different fusogenic properties and hepato-neuropathogenesis from two closely related murine-β-coronavirus strains. Using a reverse genetic system, the authors generated a new mutant virus, RSMHV2(PP), which was added a proline to the center of the fusion peptide of the spike. Based on the in vitro and in vivo comparative study between the RSMHV2(PP), RSA59 (PP) and RSMHV2 (P), they found that the insertion of the proline resulted in enhancing the virus fusogenicity, spread, and consecutive neuropathogenesis. This study is solid, experiments are well designed and support the conclusions. Some suggestions are as follows.

The resolution of Figure 1B is too low, hard to read the sequences.

Line 206 It seems the virus was given at 50% LD50 dose. The dose for RSMHV2 (P) is 100 PFU/ml, while the dose for RSMHV2 (PP) was 20,000 PFU/ml. Does this mean the addition of one proline in the FP significantly attenuated the mutant virus? Could the authors explain this?

Did the authors do viral growth curves of the RSMHV2(PP) and RSMHV2 (P) mutant viruses to see if the addition of the proline affected the viral characteristics?

Reviewer 2 Report

In this manuscript, Safiriyu et al tried to determine what exact role of the extra proline in fusion peptide between MHV A59 and MHV2 S proteins plays in viral infection and pathogenesis in mice. They constructed the RSMHV2(PP) mutant virus and performed extensive in vitro and in vivo analysis, and conclude that “these two central prolines of the FP is essential for fusogenicity and pathogenesis”. However, there are some major concerns about their experimental design and interpretation of data.

  1. MHV A59 S protein is quite different with that of MHV2, MHV A59 virus can spread to adjacent cells through pH-dependent receptor-independent syncytia formation (RIS), but MHV2 S proteins can not mediate RIS.
  2. It is furin site, not these two central prolines of the FP, which plays major role in fusogenicity and pathogenesis of MHV A59 virus.
  3. LD50 for RSMHV2(P) was about 100 pfu, whereas LD50 for RSMHV2(PP) was 20,000 pfu, indicating that the mutant virus might be attenuated. One step and multi-step growth curve should be provided and compared with RSMHV2 (P).
  4. There are about 200-fold difference in amount of viruses used in vivo between RSMHV2 (P) and RSMHV2(PP), the difference observed in the studies might result from different amount of viruses.

Round 2

Reviewer 2 Report

This reviewer has no further question.